# Adrenal Insufficiency Induced by Continued Abiraterone Acetate Use in a Prostate Cancer Patient in Remission: The Dangers of Unmonitored Long-Term Therapy Without Corticosteroids

**DOI:** 10.3390/curroncol32030156

**Published:** 2025-03-10

**Authors:** Ahmed S. Mohamed, Ahmad R. Awwad, Angel Ann Chacko, Shraboni Dey, Brianna Braithwaite, Ruchi Bhuju, Sameh Elias

**Affiliations:** 1Department of Internal Medicine, Hackensack Meridien Health, Palisades Medical Center, North Bergen, NJ 07047, USA; angel.chacko@hmhn.org (A.A.C.); shraboni.dey@hmhn.org (S.D.); brianna.braithwaite@hmhn.org (B.B.); ruchi.bhuju@hmhn.org (R.B.); 2Transitional Year Program, Hackensack Meridien Health, Palisades Medical Center, North Bergen, NJ 07047, USA; ahmad.awwad@hmhn.org; 3Internal Medicine Program, Hackensack Meridien Health, Palisades Medical Center, North Bergen, NJ 07047, USA; sameh.elias@hmhn.org

**Keywords:** abiraterone acetate, adrenal insufficiency, prostate cancer, mineralocorticoid excess syndrome, prednisolone, CYP17A1

## Abstract

This case report presents a rare occurrence of adrenal insufficiency induced by Zytiga (abiraterone acetate) in a patient with high-risk localized prostatic adenocarcinoma. Abiraterone acetate is a potent, selective and irreversible CYP17A1 inhibitor and is commonly used in the treatment of prostate cancer, but it can cause various endocrine side effects, especially if not used concurrently with the appropriate treatment. The clinical implications of this adverse event and management strategies are discussed here in this case report to raise awareness about this potential risk in patients with prostate cancer undergoing treatment with abiraterone acetate, especially when used in an erroneous manner without monitoring.

## 1. Introduction

Abiraterone acetate (Zytiga) is a second-generation (novel) antiandrogen that selectively and irreversibly inhibits the enzyme CYP17A1, which plays a key role in androgen biosynthesis inhibition in the testes, adrenal glands, and prostatic tissue. In metastatic settings, it is used with prednisone/prednisolone and androgen deprivation therapy (ADT) as a first-line treatment for high-risk disease metastatic castrate-sensitive prostate cancer (mCSPC), as it showed improved overall survival (OS) and radiographic progression-free survival (PFS) as compared to ADT alone [1,2]. An important mechanism by which androgen deprivation therapy fails to control tumor progression is that the tumor cells evade it by converting steroid precursors into androgens intracellularly. This mechanism promotes novel antiandrogen drugs, including abiraterone acetate, to be used in the treatment of metastatic castrate-resistant prostate cancer (mCRPC) before and after progression on first-line treatment (e.g., ADT+ chemotherapy (docetaxel) [3,4,5]. In the localized non-metastatic settings, long-term ADT in combination with abiraterone acetate+ prednisone for 2 years with concurrent radiotherapy is the standard of treatment in high-/very-high-risk prostate cancer (>/=Grade 4/Gleason 8-10, PSA > 40 or >T3a), which showed improved OS and failure-free survival compared ADT alone [1].

Through CYP17A1 inhibition, however, abiraterone acetate can cause cortisol deficiency with a state of excess of mineralocorticoids in a similar manner to 17-alpha hydroxylase deficiency in congenital adrenal hyperplasia (CAH). To avoid the clinical manifestations of such disorders, patients are started on prednisone simultaneously to mitigate the potential occurrence of mineralocorticoid excess syndrome (MES) and adrenal insufficiency (AI). Due to its complex mechanism of action, there are no established criteria to diagnose or confirm MES or AI. Its mechanism of action can result in a wide range of side effects, such as hypertension and hypokalemia, which are indicative of MES. On the other hand, symptoms of adrenal insufficiency, most notably fatigue, can occur. Adrenal insufficiency is a rare, potentially serious and underrecognized complication in patients receiving abiraterone acetate therapy. It tends to occur more in periods of stress, when the lack of cortisol synthesis induced by abiraterone acetate overrides the MES state. Here, we describe a case of abiraterone acetate-induced adrenal insufficiency after having a thoracic spine fracture in a patient with localized, high-risk prostate cancer who, despite achieving remission with ADT and abiraterone for 24 months, continued to take abiraterone acetate without prednisone and proper monitoring.

## 2. Case Presentation

### 2.1. Patient Information

This case study considers a 77-year-old male with a history of stage IIIC (cT2N0M0), high-risk prostate cancer (1ry Gleason score (4 + 5), stage IIIC) diagnosed in 2019, hypertension, Type II DM and dyslipidemia who presented with symptoms of failure to thrive (FTT) with a main complaint of dysphagia to solid food associated with dyspepsia and severe fatigue. The patient started developing these symptoms after sustaining a fall in early October of 2024, which resulted in an acute, non-displaced, transverse fracture of the T11 and T12 superior endplates. Ever since, the patient had been having generalized weakness and fatigue with no focal deficits. The patient denied sensory deficits, saddle anesthesia, and bladder or bowel incontinence. The patient’s dysphagia had also been worsening, forcing him to eat thinner food, because he felt pain and sensation due to food getting stuck in his lower chest after he ate. He denied nausea, vomiting, diarrhea, fever, chills, cough, shortness of breath, hemoptysis or any other symptoms. The patient was admitted to the hospital prior to that encounter for a workup of the same symptoms. He underwent upper GI endoscopy, which showed gastritis/erosions with duodenal clean-based ulcer and prominent ampulla. Biopsies were taken, which came back negative for malignancy and positive for H pylori. He was discharged on triple therapy, only to present with worsening symptoms on the same day. Regarding his prostate cancer treatment, the patient received concurrent external beam radiotherapy with long-term ADT (Lupron injection Q 3 months) in addition to abiraterone acetate and prednisolone, which should have ended in July 2022.

Patient’s blood pressure on admission: 95/60.

Lab findings:

CBC: WBCs: 3.51 × 10^3^/μL; Hgb: 11.1 g/dL; platelet count: 278 × 10^3^/μL.

Chemistry: glucose: 108 mg/dL; Na: 142 mmol/L; K: 4.2 mmol/L; Cl: 105 mmol/L; HCO3: 26 mmol/L; elevated BUN: 31 mg/dL; creatinine: 1.82 mg/dL; Mg: 1.5 mg/dL; Ca: 8.4 mg/dL; albumin: 2.8 g/dL; PO4: 2.6 mg/dL; ALT: 49 U/L; AST: 28 U/L; alkaline phosphatase: 82 U/L; lipase: 142 U/L; T PSA: <0.01 ng/mL.

CT of the chest, abdomen, and pelvis showed no evidence of metastatic disease and no evidence of acute pancreatitis.

The patient was admitted for further workup and management. After a very detailed medication reconciliation with the patient and the family, the patient was found to have been intermittently on abiraterone acetate, with days of him taking up to 12 pills a day (3000 mg) without prednisone, despite achieving remission after finishing two years of treatment and having undetectable PSA levels on follow ups.

These findings, combined with the patient’s clinical presentation, were suggestive of a possible adrenal insufficiency given the patient’s recent fracture, which might have put him under physiological stress in the context of continued abiraterone acetate use without prednisone.

### 2.2. Diagnostic Evaluation

Given the patient’s symptoms and the laboratory findings, the possibility of adrenal insufficiency induced by abiraterone acetate, especially with the erroneous intake without corticosteroids, was considered. His morning cortisol level was obtained, which was markedly low (1.4 µg/dL, normal range: 5–23 µg/dL), followed by failure to achieve peak cortisol secretion upon a cosyntropin stimulation (30 min: 9.8 and 60 min: 14.4) test, which confirmed the diagnosis of adrenal insufficiency.

### 2.3. Treatment and Management

We immediately discontinued abiraterone acetate and kept the patient on intravenous fluids to correct his volume status. We added hydrocortisone (5 mg in the morning and 5 mg in the afternoon) to replace the deficient corticosteroid levels. The patient’s blood pressure and symptoms of fatigue began to improve within 24–48 h, and he was discharged with instructions to taper the hydrocortisone dose gradually over the next few weeks. The patient was also given zoledronic acid during his hospital course and was discharged to a subacute rehab center. Neurosurgery was consulted on during the hospital stay, and conservative management was recommended for his T11 and T12 fractures. The patient was instructed to stop abiraterone acetate indefinitely, as it should have been stopped in 2022 after 2 years in combination with ADT as per the STAMPEDE trial [1].

## 3. Discussion

Adrenal insufficiency induced by abiraterone acetate is a rare but serious adverse event that arises due to the drug’s inhibition of CYP17A1, which leads to a reduction in cortisol production. This can result in primary adrenal insufficiency. This is why it is always given with prednisone/prednisolone [1,2,4,5]. The most likely pathophysiology involves the suppression of adrenal gland function due to a lack of necessary precursors for cortisol synthesis, resulting in impaired steroidogenesis and an inability to reach an adequate glucocorticoid response, especially in times of stress.

CYP17 inhibition can result in high ACTH because of inhibited steroidogenesis, which can lead to the formation of excess precursors upstream of CYP17, resulting in mineralocorticoid excess syndrome (MES). This pathogenesis is similar to that of 17 alpha hydroxylase deficiency, which is a CYP17A1 in congenital adrenal hyperplasia (CAH), in which an upstream substrate, corticosterone, accumulates and also acts as a weak glucocorticoid and, sometimes, at very high levels, suppresses ACTH (Figure 1) [6]. This mineralocorticoid excess, together with the weak glucocorticoid activity, usually mitigates the clinical consequences of adrenal insufficiency [7]. This parallel between abiraterone acetate and CAH led the investigators’ attention away from the potential of abiraterone acetate to induce adrenal insufficiency in early trials that utilized abiraterone acetate in the treatment of prostate cancer. In the initial phase I and II trials, abiraterone acetate was administered with and without glucocorticoid therapy in castrate and non-castrate prostate cancer patients. In one trial that monitored cortisol level and performed cosyntropin testing in patients receiving abiraterone acetate, it was found that although there was no overt adrenal insufficiency, a reduction in cortisol levels and a suboptimal response to the cosyntropin stimulation test were observed in a few patients [8]. In another phase I trial that evaluated abiraterone acetate in patients with castrate-resistant prostate cancer who were previously on ketoconazole, MES occurred, which was treated with an aldosterone receptor antagonist [9]; however, the adjunct glucocorticoid therapy also had the additional effect of improving tumor response, as well as ameliorating the incidence of MES occurrence [10,11]. These findings established the role of glucocorticoid coadministration in phase III trials. Prednisone 10 mg once a day was given concurrently with abiraterone in COU-AA-301/302, while prednisone 5 mg was given in the STAMPEDE and LATITUDE trials [1,2,4,5]. Despite the drastically decreased incidence of MES occurrence, fatigue, which was queried to be indicative of AI, was the most common side effect.

Upon a review of the literature, we found two published case studies reporting AI secondary to abiraterone use. Baloch et al. (Table 1) reported two cases of metastatic prostate adenocarcinoma who received abiraterone acetate with concurrent prednisone, one of whom developed MES. The other patient was a patient with stage IV metastatic prostatic adenocarcinoma who was treated with 1000 mg/day with prednisone 5 mg/day. The patient was admitted for the workup of fatigue and shortness of breath, 2 weeks after being treated for septic shock in the ICU. The patient was fully worked up and was negative for the progression of his cancer, HIV. The full immunological and infectious workup came back negative. AI was suspected by the authors, and diagnosis was confirmed by AM cortisol and cosyntropin stimulation tests. The patient’s prednisone dose was increased to 20 mg with an improvement in symptoms. Following discharge, the patient was weaned off to 10 mg (5 mg twice a day) with a stable condition. Further weaning was not successful due to symptom recurrence. The patient was then further tested for AM cortisol level while being off prednisone for 24 h, which came back low, despite clinical improvement, and the patient was therefore continued on prednisone 5 mg twice a day [12]. Another distinct study (Table 1) was a case series of three patients, two of which developed adrenal insufficiency secondary to abiraterone acetate. The first patient, diagnosed with high-risk/volume mCSPC with widespread osseous lesions, was on degarlix, abiraterone 1000 mg and prednisone 5 mg and presented with fatigue, shortness of breath, hypotension and atrial fibrillation with rapid ventricular response. The patient’s blood pressure did not improve despite appropriate hydration and cardioversion. Abiraterone was not administered during the illness. Adrenal insufficiency was confirmed by their AM cortisol level, ACTH and inadequate response to cosyntropin stimulation. The patient’s prednisone dose was doubled to 10 mg for a few days, abiraterone was stopped, and the patient was instructed to continue prednisone 5 mg once daily for the diagnosis of drug-induced adrenal insufficiency. The second reported patient had mCRPC who progressed on ADT and was started on abiraterone acetate and prednisone. A few months later, he presented with FTT and abdominal pain. Abiraterone and prednisone were stopped for 8 days, and after said period, the patient continued to experience hypoglycemia and hypotension, which were unresponsive to fluids. AI was suspected and confirmed by the patient’s AM cortisol level and inability to achieve response to cosyntropin stimulation. Abiraterone was stopped, and the patient was discharged on hydrocortisone to be tapered and then kept on a physiological dose of hydrocortisone for AI [11].

Symptoms of adrenal insufficiency can be nonspecific and may include fatigue, hypotension, dizziness, weight loss and electrolyte disturbances such as hyperkalemia and hyponatremia. As seen in our very atypical case of a patient with localized prostate cancer who underwent 5 years in remission with the anticipation of them being taken off treatment, meticulous medication review and the early identification of symptoms are important. Fatigue, hypotension and dysphagia leading up to FTT were the side effects seen in our patient. Although electrolyte abnormalities, including hyponatremia, hyperkalemia and hypoglycemia, are typical hallmarks of adrenal insufficiency, none were reported in our case. This can be attributed to the complex mechanism of abiraterone acetate. Fatigue, however, together with hypotension, raised our suspicion towards the possibility of AI. Based on the literature review of the aforementioned case studies and our case presentation, we believe that despite the state of MES that abiraterone acetate creates, AI can very rarely occur, especially in severe periods of physiological stress, when the lack of cortisol necessary to reach a physiological response overrides the weak mineralocorticoid activity of the accumulated glucocorticoids upstream of the inhibited CYP17A1 without or even with the coadministration of concomitant prednisone. This might explain the reason why for patients who receive active treatment with abiraterone acetate and corticosteroids, it is still a challenge to pinpoint the optimal dose of adjunct corticosteroids necessary to achieve the right balance between the MES state and the potential development of AI. However, the FDA label recommends the use of prednisone 10 mg with abiraterone in mCRPC as per the COU-AA-301/302 study design [4,5] and prednisone 5 mg/prednisolone (STAMPEDE) in mCSPC as per the STAMPEDE and LATITIUDE study designs [1,2]. There still remains no evidence to support which is superior.

Another possible reason why our patient exhibited these symptoms could be attributed to withdrawal from chronic prednisone use after achieving remission, particularly in the absence of MES symptoms, which are typically associated with the ongoing effects of abiraterone acetate alone. Considering the complexity of how abiraterone acetate works, as well as the atypical presentation that differs from the aforementioned case studies reporting AI in patients with active disease who received both abiraterone and concurrent prednisone, we believe that, in lieu of the clinical picture our patient presented with, his findings may be attributed to one or both factors.

For our patient with high-risk localized prostate cancer who underwent 5 years in remission and had T PSA < 0.1, as checked during hospitalization, we discontinued abiraterone acetate and initiated corticosteroid replacement therapy with hydrocortisone, which improved his symptoms drastically

## 4. Conclusions

This case highlights the potential for abiraterone acetate to induce adrenal insufficiency, a serious but treatable side effect, especially at times of stress and when administered in an incorrect manner without concurrent corticosteroids. Adrenal insufficiency can also still occur in patients receiving abiraterone with concurrent corticosteroids. Currently, there is no evidence to support whether the 5 mg or 10 mg dose of prednisone is superior in patients receiving active treatment with abiraterone. When choosing, one must weigh the risk of abiraterone’s possible side effects of AI with the adverse events related to hypercortisolism. However, there is evidence to support both. Non-oncology clinicians should also be aware of this risk, particularly in patients presenting with unexplained, nonspecific symptoms, particularly fatigue, hypotension and failure to thrive, and with no clear indication to be on such medication. They should carry out a thorough review of patients’ medications. Early diagnosis and prompt treatment with corticosteroid replacement therapy are crucial. Further research is still needed to better understand the mechanisms and incidence of adrenal insufficiency in patients treated with abiraterone acetate.

## Figures and Tables

**Figure 1 curroncol-32-00156-f001:**
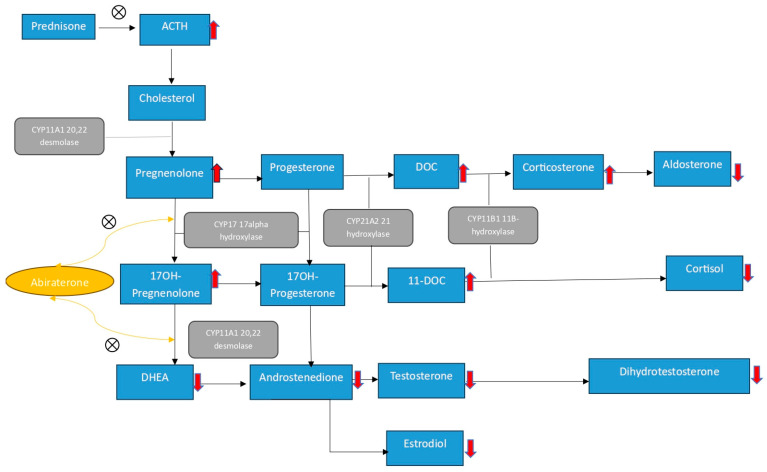
Abiraterone acetate inhibits 17-alpha hydroxylase and 17, 20-lyase, further decreasing the downstream synthesis of cortisol, leading to an ACTH surge with the accumulation of upstream CYP17 mineralocorticoids, which have weak glucocorticoid activity. Prednisone decreases this compensatory ACTH surge and thus decreases the state of MES. Abbreviations: ACTH: Adrenocorticotrophic hormone, DHEA: Dehydroepiandrosterone, DOC: Deoxycorticosterone.

**Table 1 curroncol-32-00156-t001:** Literature review of prior case studies reporting AI in patients receiving abiraterone acetate.

Baloch et al. [12]	Metastatic Prostate Cancer	Abiraterone and Prednisone 5 mg	Fatigue and Shortness of Breath	AI	Abiraterone Stopped. Prednisone Continued at 20 mg Initially and Then 10 mg
Decamps S et al. [11]	1—mCSPC 2—mCRPC	1—Degralix, abiraterone and prednisone 5 mg 2—Abiraterone and prednisone 5 mg	1—Fatigue, SOB, hypotension and atrial fibrillation 2—FTT and abdominal pain	1—AI 2—AI	1—Abiraterone stopped. Prednisone doubled to 10 mg and then continued at 5 mg 2—Abiraterone and prednisone stopped. Hydrocortisone started, tapered down and continued on physiological dose

## Data Availability

The authors confirm that the data supporting the findings of this study are available within the article.

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
