# Peer review of "Adrenal Insufficiency Induced by Continued Abiraterone Acetate Use in a Prostate Cancer Patient in Remission: The Dangers of Unmonitored Long-Term Therapy Without Corticosteroids"

_curroncol, 2025, doi:10.3390/curroncol32030156_

Round 1
Reviewer 1 Report
Comments and Suggestions for Authors
Good case report, but it should be improved in the following aspects:
The omission of documentation for informed consent and ethical committee approval is a significant shortcoming, as these elements are essential for ensuring transparency and ethical validity in case reports.
Although the case emphasizes the importance of concomitant corticosteroid use with abiraterone, it does not offer substantial novelty.
The description of the pathophysiological mechanisms is somewhat superficial. A more detailed discussion of how CYP17A1 inhibition leads to reduced cortisol synthesis—thereby triggering a compensatory increase in ACTH and the subsequent accumulation of precursors that result in elevated mineralocorticoid levels—would improve the understanding of the underlying basis for adrenal insufficiency when abiraterone is administered without corticosteroid support.
The discussion could be further expanded to elucidate how the administration of abiraterone without adequate corticosteroid support specifically precipitates adrenal insufficiency. This should include an analysis of hormonal regulation, the role of ACTH feedback, and the potential secondary clinical effects.
Additionally, the manuscript lacks the units of measurement for several laboratory parameters. For example: CBC: WBCs: 3.51. Hgb: 11.1, Platelet count: 278 Chemistry: Glucose: 108, Na: 142, K: 4.2, Cl: 105, HCO3: 26, Elevated BUN: 31 and creatinine: 1.82, Mg: 1.5, Ca: 8.4, Albumin: 2.8, PO4: 2.6, ALT: 49, AST: 28, Alkaline phosphatase: 82, Lipase: elevated at 142, T PSA: <0.01.
The discussion should also address the electrolyte parameters observed in the case. The patient’s sodium (Na: 142 mmol/L) and potassium (K: 4.2 mmol/L) levels are within normal limits, which contrasts with the classical presentation of primary adrenal insufficiency, typically characterized by hyponatremia and hyperkalemia. This discrepancy should be discussed in the context of abiraterone-induced adrenal insufficiency.
Author Response
- The omission of documentation for informed consent and ethical committee approval is a significant shortcoming, as these elements are essential for ensuring transparency and ethical validity in case Reports.
Response: Thanks for pointing that out. Both Informed consent and ethical committee statement were obtained prior to submitting the paper. We were unaware of the journal’s policy on when and where to include as well as the wording. We also complied with the informed consent version that the journal desires, corresponded with the patient again and obtained a consent in the wording that suits the journal.
- Although the case emphasizes the importance of concomitant corticosteroid use with abiraterone, it does not offer substantial novelty. The description of the pathophysiological mechanisms is somewhat superficial. A more detailed discussion of how CYP17A1 inhibition leads to reduced cortisol synthesis—thereby triggering a compensatory increase in ACTH and the subsequent accumulation of precursors that result in elevated mineralocorticoid levels—would improve the understanding of the underlying basis for adrenal insufficiency when abiraterone is administered without corticosteroid support. The discussion could be further expanded to elucidate how the administration of abiraterone without adequate corticosteroid support specifically precipitates adrenal insufficiency. This should include an analysis of hormonal regulation, the role of ACTH feedback, and the potential secondary clinical effects.
Response:
A thorough explanation of how CYP17A1 inhibition results in MES and how adrenal insufficiency could be induced is discussed here with an illustrative graph. We believe that although MES occurs when CYP17A1 is inhibited as a result of increased ACTH and upstream mineralocorticoids that have weak glucocorticoid effect, AI can very rarely occur if the true cortisol deficiency overrides the MES state
we tried to explain the reason why we think AI occured in the context of the patient's use of abiraterone, supported by the previous case reports as well as the lack of evidence to support what the optimal dose of prednisone is in the context of active disease.
- Additionally, the manuscript lacks the units of measurement for several laboratory parameters. For example: CBC: WBCs: 3.51. Hgb: 11.1, Platelet count: 278 Chemistry: Glucose: 108, Na: 142, K: 4.2, Cl: 105, HCO3: 26, Elevated BUN: 31 and creatinine: 1.82, Mg: 1.5, Ca: 8.4, Albumin: 2.8, PO4: 2.6, ALT: 49, AST: 28, Alkaline phosphatase: 82, Lipase: elevated at 142, T PSA:0.01.
Response: That was addressed. Refer to the revised manuscript
- The discussion should also address the electrolyte parameters observed in the case. The patient’s sodium (Na: 142 mmol/L) and potassium (K: 4.2 mmol/L) levels are within normal limits, which contrasts with the classical presentation of primary adrenal insufficiency, typically characterized by hyponatremia and hyperkalemia. This discrepancy should be discussed in the context of abiraterone-induced adrenal insufficiency.
Response: Although electrolyte disturbances are classic of AI, there was none in our case study. Fatigue and borderline low BP were the only symptoms. That might be due to the complex action of abiraterone acetate.

Reviewer 2 Report
Comments and Suggestions for Authors
Authors present a case of adrenal insufficiency in a patient who had been treated with 2 years of ADT + abiraterone/prednisone who continued to take abiraterone without prednisone. Adrenal insufficiency and mineralocorticoid excess syndrome are known toxicities of abiraterone and why prednisone is co-administered with the abiraterone. This case highlights the risk of AI occuring, especially in context of prednisone withdrawal. The authors highlight that the lack of monitoring and co-administered prednisone was responsible. Interestingly, the labs (and lack of hypertensoin) were NOT consistent with mineralocorticoid excess syndrome but instead with primary adrenal insufficiency. It is important to add to this report that the cause may been from the prednisone withdrawal and chronic prednisone use rather than purely abiraterone.
Comments:
- Line 33: correct to "high risk" prostate cancer, as LATITUDE, which is referenced, was not exclusively in high volume disease. The FDA approvals are not restricted to high volume disease.
- Line 39: it is approved for before and after chemotherapy, the citation (5) is prior to chemotherapy
- Given the lack of mineralocorticoid excess syndrome, it seems as likely that the AI was due to prior chronic corticosteroid use with recent discontinuation rather than direct ongoing abiraterone effect. I would suggest adding this possibility to the discussion.
- A couple of grammatical clarifications needed
minor editing needed
Author Response
-
- Line 33: correct to "high risk” prostate cancer, as LATITUDE, which is referenced, was not exclusively in high volume disease. The FDA approvals are not restricted to high volume disease.
Respone: Corrected. Thanks for pointing that out
- Line 39: it is approved for before and after chemotherapy, the citation (5) is prior to chemotherapy
Response: Corrected
- Given the lack of mineralocorticoid excess syndrome, it seems as likely that the AI was due to prior chronic corticosteroid use with recent discontinuation rather than direct ongoing abiraterone effect. I would suggest adding this possibility to the discussion.
Response: Added that as a possible explanation for the patient's presentation. We also added that the presentation can be attributed to eboth mechanisms and although abiraterone acetate alone should create a state of MES, we believe that in periods of physiological stress, the true cortisol deficiency overrides the MES.

Round 2
Reviewer 1 Report
Comments and Suggestions for Authors
The authors appropriately addressed all the questions, and the manuscript has substantially improved. Just a small detail: the affiliations are incomplete, as the city and country are missing.
Comments on the Quality of English LanguageNo comments